# Antimicrobial resistance of rapidly growing mycobacteria isolated from companion animals in Taiwan

Shu-Wen Chen,[1] Ter-Hsin Chen,[1,2] Wei-Hsiang Huang,[3] Chia-Chun Hou,[4,5] Chen-Jou Lin,[5,6,7] Yi-Fu Chang,[7,8] Hsin-Yi Wu,[9] Ying-Chen Wu[1,2]

**ABSTRACT** Rapidly growing mycobacteria (RGM) are omnipresent nontuberculous mycobacteria that cause opportunistic infections in animals and humans. Without knowledge of the epidemiology and antimicrobial susceptibility of RGM in companion animals in Taiwan, diagnostic and therapeutic regimens are limited. To address this, we collected 44 RGM isolates from 25 dogs and 19 cats from 2018 to 2021 and investigated their antimicrobial susceptibility and macrolide-resistance genes. The most prevalent RGM were *Mycobacterium fortuitum* complex (MFC), accounting for 20 isolates (14 dogs and 6 cats), and *M. abscessus* complex (MABC), accounting for 20 isolates (9 dogs and 11 cats). More than 80% of the RGM isolates were susceptible to linezolid and amikacin. All MABC isolates were resistant to at least three groups of essential antibiotics, including tetracyclines, fluoroquinolones, and trimethoprim-sulfamethoxazole, whereas 75% of MABC isolates were susceptible to clarithromycin. In contrast, 35% of MFC isolates were susceptible to clarithromycin, but these isolates varied in resistance to other antibiotics. The presence of inducible macrolide resistance was further confirmed by the coherence between the minimum inhibitory concentrations of clarithromycin and the presence of *erm* genes. In conclusion, our results showed that MABC and MFC are the major pathogens causing RGM infections in dogs and cats. The variability in their antimicrobial susceptibility profiles makes treatment challenging, particularly with the development of inducible resistance to macrolides. Local epidemiological data and comprehensive microbiological examinations are critical for diagnosis and treatment planning, whereas resistance gene detection aids in the rapid evaluation of RGM resistance to macrolides.

**IMPORTANCE** Rapidly growing mycobacteria (RGM) are opportunistic pathogens in both humans and animals, posing significant challenges in diagnosis and treatment. The variable antimicrobial resistance profiles and inducible macrolide resistance complicate the design of multidrug regimens. Research on RGM infections in dogs and cats is limited, particularly studies examining inducible macrolide resistance. This study identified *Mycobacterium abscessus* complex and *M. fortuitum* complex as the predominant species in dogs and cats in Taiwan. Both species exhibited poor susceptibility to many antibiotics. *M. fortuitum* demonstrated lower minimum inhibitory concentration (MIC) values for fluoroquinolones and higher MIC values for clarithromycin, whereas *M. abscessus* complex showed the reverse pattern. Inducible macrolide resistance was present in our RGM isolates, and the detection of the *erm* genes provided a reliable prediction. These results support clinical diagnosis and the formulation of multidrug treatment regimens for RGM infections in dogs and cats.

**KEYWORDS** rapidly growing mycobacteria, nontuberculous mycobacteria, antibiotic resistant, dog, cat, animal

**Peer Reviewer** Yung Chun Chen, Taichung Veterans General Hospital, Taichung, Taiwan

Address correspondence to Ying-Chen Wu, pata0112@gmail.com.

The authors declare no conflict of interest.

Rapidly growing mycobacteria (RGM) are ubiquitous opportunistic pathogens present in water, soil, and aerosols. RGM are gram-positive, acid-fast bacteria characterized by the formation of colonies within 7 days of incubation in clinical specimens. Common pathogenic species of RGM include *Mycobacterium abscessus* complex (MABC), *Mycobacterium chelonae*, *Mycobacterium fortuitum* complex (MFC), and *Mycobacterium smegmatis* (1, 2). In companion animals, RGM infections mostly result from percutaneous trauma such as bites or surgery and can cause chronic non-healing dermatitis, cellulitis, or panniculitis. In addition to skin lesions, RGM cause pyogranulomatous pneumonia, lymphadenitis, and systemic infections in both immunocompromised and healthy individuals (3–7). The treatment of RGM infections is challenging, and there are differences in antimicrobial resistance among species (1, 8–11). MABC is a group of RGM that includes *M. abscessus*, *M. bolletii*, and *M. massiliense* (12). MABC is resistant to most anti-mycobacterial agents and other antimicrobials, thus leading to treatment difficulties (1, 8, 13–15). Macrolides are the major therapeutic agents used for MABC (8–10, 16, 17). Macrolides, including azithromycin and clarithromycin, inhibit bacterial protein synthesis by binding to the 23S subunit of the 50S rRNA. Methylation of the peptidyltransferase region in the 23S subunit by a methyltransferase reduces the affinity between the macrolide and target region, resulting in inducible resistance that leads to an increased minimum inhibitory concentration (MIC) after exposure to macrolides (18, 19). The *erm* genes confer resistance to MFC and *M. smegmatis* (19, 20), whereas *erm*(41) confers resistance to macrolides in MABC (21). The large deletion of the *erm*(41) sequence led to a 397 bp polymerase chain reaction product in *M. massiliense*, resulting in a non-functional methyltransferase and no inducible resistance to clarithromycin. Another non-functional *erm*(41) gene is caused by a T→C mutation at nucleotide 28, which has been found in both *M. abscessus* and *M. bolletii* (21, 22). Recently, a plasmid-mediated macrolide resistance gene, *erm*(55), was shown to confer inducible resistance in *M. chelonae*, highlighting the importance of testing for inducible macrolide resistance in *M. chelonae* (23, 24).

Another macrolide-resistance gene, *rrl*, encodes the peptidyltransferase region of the 23S rRNA gene, and the A→G/C point mutation at *rrl* position 2058, 2059, or the adjacent positions modifies the structure of the 23S subunit and results in acquired resistance (25–27). Amikacin is a critical antibiotic for RGM infections and is considered a suitable parenteral therapeutic option for multidrug therapies (8–11, 16, 28). Acquired aminoglycoside resistance in RGM, primarily due to modifications in the 30S ribosomal subunit (29, 30), complicates treatment strategies.

Because of the absence of epidemiological data and the presence of multidrug resistance, RGM infections in dogs and cats pose predicaments in diagnosis and therapy. This study was performed to understand the species distribution and resistance patterns of RGM isolated from companion animals, to help address these clinical issues and raise public health concerns regarding the antimicrobial resistance of this category of zoonotic pathogens.

## RESULTS

Among the 42,433 clinical specimens analyzed, we confirmed 44 RGM isolates from 25 dogs and 19 cats. Among these patients, 20 isolates of MABC (9 dogs and 11 cats), 20 isolates of MFC (14 dogs and 6 cats), 1 isolate of *M. chelonae*, 1 isolate of *M. nivoides*, and 2 isolates of *M. smegmatis* were identified. The number of isolates and species identified at each site is listed in Table 1. The identified MABC isolates included *M. abscessus, M. bolletii*, and *M. massiliense*. Among the dogs and cats, *M. massiliense* was the most common subspecies. The MFC isolates identified included *M. fortuitum*, *M. houstonense*, *M. mageritense*, *M. mucogenicum*, *M. peregrinum*, and *M. porcinum* (1, 31). In dogs, *M. fortuitum* was the most common species within the MFC group, whereas in cats, there were no dominant MFC species. Detailed information about each patient is provided in Table S1.

**TABLE 1** Number of clinical isolates of different species by isolation site[d]

| Species | No. of isolates (dog/cat) | Sites of isolation: dog/cat | | | |
|---|---|---|---|---|---|
| | | Cutaneous (dog/cat) | Respiratory (dog /cat) | Genitourinary (dog /cat) | Lymph node (dog/cat) |
| **M. abscessus complex** | **9/11** | **9/5** | **0/5** | **0/1** | |
| M. abscessus | 2/4 | 2/2 | 0/1 | 0/1[a] | |
| M. bolletii | 1/1 | 1/0 | 0/1 | | |
| M. massiliense | 6/6 | 6/3 | 0/3 | | |
| **M. fortuitum complex** | **14/6** | **11/1** | **0/5** | **1/0** | **2/0** |
| M. fortuitum | 11/2 | 10/1 | 0/1 | | 1/0 |
| M. peregrinum | 1/0 | 1/0 | | | |
| M. mageritense | 1/1 | | 0/1 | | 1/0 |
| M. porcinum | 1/0 | | | 1[b]/0 | |
| M. mucogenicum | 0/2 | | 0/2 | | |
| M. houstonense | 0/1 | | 0/1 | | |
| **M. nivoides** | **1/0** | **1/0** | | | |
| **M. chelonae** | **0/1** | | **0/1** | | |
| **M. smegmatis** | **1/1** | **1/1** | | | |
| Total[c] | 25/19 | 21/6 | 1/12 | 1/1 | 2/0 |

[a]Isolated from the urine sample collected via cystocentesis.
[b]Isolated from the prostate collected via fine needle aspiration.
[c]The culture results were obtained from 44 patients (25 dogs and 19 cats). During the treatment period, follow-up examinations were conducted on 11 patients, and the results of additional isolates from these patients are presented in Table S2.
[d]The values in bold indicate individual species or complexes (non-bold text denotes a subcategory within a complex).

To compare and clarify the antimicrobial resistance patterns across different species, the MIC values of the RGM isolates are shown in Tables 2–4. More than 80% of MABC and MFC isolates were susceptible to amikacin and linezolid. All MABC isolates were resistant to trimethoprim-sulfamethoxazole, doxycycline, ciprofloxacin, moxifloxacin, enrofloxacin, and levofloxacin. Susceptibility to cefoxitin and imipenem varied. Clarithromycin resistance in MABC isolates was observed in 11.1% of the dogs (1 out of 9) and 36.4% of the cats (4 out of 11) (Tables 2 and 5). The MFC isolates exhibited lower MIC values for fluoroquinolones and imipenem, while generally demonstrating high MIC values and variable resistance to clarithromycin, doxycycline, cefoxitin, and trimethoprim-sulfamethoxazole (Table 3).

The results of the presence of erm(41) and macrolide resistance are shown in Table 5. Among the 20 MABC isolates, all of the *M. massiliense* isolates ($n$ = 12) had a truncated 246 bp erm(41), whereas all *M. bolletii* isolates ($n$ = 2) and one of the six *M. abscessus* isolates (CMAB-03) had a 522 bp erm(41) with the T28C mutation. In these isolates, the MICs of clarithromycin remained <2 µg/mL after 14 days of incubation. The remaining five *M. abscessus* isolates had a nontruncated 522 bp erm(41), and these isolates exhibited MICs > 16 µg/mL. For the MICs of azithromycin after 14 days of incubation, only two isolates (*M. massiliense*, FMAB-04 and FMAB-12, with the 246 bp erm(41)) showed MICs < 4 µg/mL, while the remaining 18 isolates exhibited MICs ranging from 4 to >16 µg/mL, even though 13 of these isolates (10 *M. massiliense*, 1 *M. abscessus*, and 2 *M. bolletii*) carried non-functional erm(41) genes.

MFC isolates (susceptibility rate: 35.0%, 7/20) exhibited reduced susceptibility to clarithromycin compared with MABC isolates (susceptibility rate: 75.0%, 15/20). The results of erm detection in MFC, *M. nivoides*, *M. smegmatis*, and *M. chelonae* are presented in Table 6. For the MICs of clarithromycin after 14 days of incubation, among the 20 MFC isolates, 76.9% (10/13) of *M. fortuitum* isolates and 42.9% (3/7) of other MFC isolates tested positive for the erm and showed MIC values > 2 µg/mL. For the erm-negative isolates, the MICs of clarithromycin remained <4 µg/mL. For the MICs of azithromycin, all MFC isolates exhibited MICs ranging from 4 to >16 µg/mL after 14 days of incubation, regardless of whether they carried erm genes.

**TABLE 2** Antimicrobial susceptibilities of *Mycobacterium abscessus* complex isolates ($n$ = 20)[a]

| Antimicrobial agents | MIC (µg/mL)[d] | | | % Susceptibility[e] | | |
|---|---|---|---|---|---|---|
| | Range | MIC$_{50}$ | MIC$_{90}$ | Total | Dog MABC (ab/bo/ma) | Cat MABC (ab/bo/ma) |
| Amikacin[b] | <8−32 | 8 | 32 | 85.0 | 88.9 (100/100/83.3) | 81.8 (100/100/66.7) |
| Gentamicin[b] | 4−32 | 16 | 16 | 15.0 | 11.1 (0/100/0) | 18.2 (0/100/16.7) |
| Azithromycin[b,c] | 2−>16 | 8 | >16 | 10.0 | 0 (0/0/0) | 18.2 (0/0/33.3) |
| Clarithromycin[c] | <1−>16 | <1 | >16 | 75.0 | 88.9 (50.0/100/100) | 63.6 (0/100/100) |
| Cefoxitin | 16−64 | 32 | 32 | 45.0 | 55.6 (50.0/0/66.7) | 36.4 (25.0/100/33.3) |
| Imipenem | 4−16 | 8 | 8 | 15.0 | 0 (0/0/0) | 27.3 (50.0/0/16.7) |
| Enrofloxacin[b] | 4−>4 | >4 | >4 | 0 | 0 (0/0/0) | 0 (0/0/0) |
| Ciprofloxacin | 4−>4 | >4 | >4 | 0 | 0 (0/0/0) | 0 (0/0/0) |
| Levofloxacin[b] | 4−16 | 8 | 16 | 0 | 0 (0/0/0) | 0 (0/0/0) |
| Moxifloxacin | 8−>8 | >8 | >8 | 0 | 0 (0/0/0) | 0 (0/0/0) |
| Doxycycline | >16 | >16 | >16 | 0 | 0 (0/0/0) | 0 (0/0/0) |
| Linezolid | <4−16 | 4 | 16 | 85.0 | 100 (100/100/100) | 72.7 (50.0/100/83.3) |
| SXT | 4/76−>8/152 | >8/152 | >8/152 | 0 | 0 (0/0/0) | 0 (0/0/0) |
| AMC | >4/2 | >4/2 | >4/2 | 0 | 0 (0/0/0) | 0 (0/0/0) |

[a]ab, *M. abscessus*; AMC, amoxicillin-clavulanate; bo, *M. bolletii*; ma, *M. massiliense*; MABC, *Mycobacterium abscessus* complex; MIC, minimum inhibitory concentration; SXT, trimethoprim-sulfamethoxazole.
[b]CLSI M24, 2018 has not addressed the breakpoints of gentamicin, enrofloxacin, levofloxacin, azithromycin, and AMC for RGM. The breakpoints were based on the interpretation criteria for *Staphylococcus* spp. (CLSI M100 and CLSI VET08).
[c]The results of azithromycin and clarithromycin are presented from a 14-day incubation.
[d]The MIC results (range, MIC$_{50}$, and MIC$_{90}$) were determined for all MABC isolates ($n$ = 20; 9 from dogs and 11 from cats).
[e]Susceptibility rates (%) were assessed for the total MABC isolates ($n$ = 20), as well as separately for Dog MABC ($n$ = 9; 2 *M. abscessus*, 1 *M. bolletii*, and 6 *M. massiliense*), and Cat MABC ($n$ = 11; 4 *M. abscessus*, 1 *M. bolletii*, and 6 *M. massiliense*).

The point mutation (A2058G/C and A2059G/C) in the *rrl* gene and the point mutations (positions 1405–1409 and 1491–1496) in the *rrs* gene were not detected in any of the 44 isolates. These results are presented in Tables S1 and S2. Of the 44 patients included in this study, 11 underwent follow-up examinations during treatment, yielding 16 additional RGM isolates. Changes in the antimicrobial resistance and associated resistance genes of these isolates during this period are detailed in Table S2.

**TABLE 3** Antimicrobial susceptibilities of *Mycobacterium fortuitum* isolates ($n$ = 13)[a]

| Antimicrobial agents | MIC (µg/mL)[d] | | | % Susceptibility[e] | | |
|---|---|---|---|---|---|---|
| | Range | MIC$_{50}$ | MIC$_{90}$ | Total | Dog | Cat |
| Amikacin | <8−8 | <8 | 8 | 100 | 100 | 100 |
| Gentamicin[b] | <4−16 | 8 | 16 | 46.2 | 54.5 | 0 |
| Azithromycin[b,c] | >16 | >16 | >16 | 0 | 0 | 0 |
| Clarithromycin[c] | 2−>16 | >16 | >16 | 23.1 | 27.3 | 0 |
| Cefoxitin | 16−128 | 32 | 128 | 23.1 | 27.3 | 0 |
| Imipenem | <4−8 | <4 | 8 | 76.9 | 72.7 | 100 |
| Enrofloxacin[b] | <0.25−>4 | 0.5 | >4 | 76.9 | 81.8 | 50 |
| Ciprofloxacin | <0.25−>4 | 0.5 | 4 | 84.6 | 90.9 | 50 |
| Levofloxacin[b] | <1−4 | <1 | 4 | 84.6 | 90.9 | 50 |
| Moxifloxacin | <0.5−4 | 0.5 | 2 | 84.6 | 90.9 | 50 |
| Doxycycline | 1−>16 | >16 | >16 | 23.1 | 27.3 | 0 |
| Linezolid | <4−16 | 8 | 16 | 84.6 | 81.8 | 100 |
| SXT | <0.5/9.5−>8/152 | >8/152 | >8/152 | 15.4 | 9.1 | 50 |
| AMC[a] | 4−>4 | >4/2 | >4/2 | 0 | 0 | 0 |

[a]AMC, amoxicillin-clavulanate; MIC, minimum inhibitory concentration; SXT, trimethoprim-sulfamethoxazole.
[b]CLSI M24, 2018 has not addressed the breakpoints of gentamicin, enrofloxacin, levofloxacin, azithromycin, and AMC for RGM. The breakpoints were referred to the interpretation criteria for *Staphylococcus* spp. (CLSI M100 and CLSI VET08).
[c]The results of azithromycin and clarithromycin are presented from a 14-day incubation.
[d]The MIC results (range, MIC$_{50}$, and MIC$_{90}$) were determined for all *Mycobacterium fortuitum* isolates ($n$ = 13; 11 from dogs and 2 from cats).
[e]Susceptibility rates (%) were assessed for the total *M. fortuitum* isolates ($n$ = 13), as well as separately for Dog *M. fortuitum* ($n$ = 11) , and Cat *M. fortuitum* ($n$ =2).

**TABLE 4** MICs of *M. smegmatis*, *M. chelonae*, *M. nivoides*, and *M. fortuitum* complex (except for *M. fortuitum*) isolates[a]

| Species | Isolate no.[b] | AN | GM | AZM | CLR | FOX | IPM | ENR | CIP | LEV | MOX | DOX | LZD | SXT | AMC |
|---|---|---|---|---|---|---|---|---|---|---|---|---|---|---|---|
| *M. smegmatis* | CSME-1 | <8 | **8** | **8** | **4** | <8 | <4 | <0.25 | <0.25 | <1 | <0.5 | 1 | <4 | **>8** | 0.25/0.125 |
| | FSME-1 | <8 | <4 | **>16** | **>16** | <8 | <4 | 0.25 | <0.25 | <1 | <0.5 | <0.5 | <4 | <0.5 | 2/1 |
| *M. chelonae* | FMCH-1 | <8 | <4 | **16** | <1 | 16 | **8** | 0.5 | **2** | <1 | 0.5 | **>16** | <4 | 1 | >4/2 |
| *M. nivoides* | CMN-1 | <8 | **8** | <1 | <1 | 8 | <4 | **>4** | <0.25 | <1 | <0.5 | <0.5 | <4 | <0.5 | 4/2 |
| *M. mucogenicum* | FMFCS-3 | **32** | **16** | **>16** | 2 | 16 | <4 | **2** | **4** | **2** | **4** | **>16** | 8 | 0.5 | >4/2 |
| | FMFCS-2 | 8 | **16** | **>16** | 1 | 16 | <4 | 0.5 | 1 | <1 | 1 | **>16** | <4 | <0.5 | >4/2 |
| *M. mageritense* | CMFCS-2 | **128** | **>64** | **>16** | **>16** | **32** | **8** | 0.5 | 0.25 | <1 | <0.5 | **>16** | <4 | **>8** | >4/2 |
| | FMFCS-1 | **128** | **>64** | **>16** | **>16** | **32** | 4 | 0.25 | 0.25 | <1 | <0.5 | **>16** | <4 | **>8** | >4/2 |
| *M. houstonense* | FMFCS-4 | <8 | 4 | **>16** | 2 | 16 | <4 | **1** | 1 | 1 | 1 | 1 | **16** | **8** | >4/2 |
| *M. porcinum* | CMFCS-3 | <8 | <4 | **>16** | **>16** | **32** | <4 | 0.5 | 0.5 | <1 | 0.5 | **>16** | <4 | **8** | 4/2 |
| *M. peregrinum* | CMFCS-1 | 8 | <4 | **4** | <1 | **32** | **8** | **>4** | **4** | **8** | **>8** | **>16** | <4 | **>8** | >4/2 |

[a]AMC, amoxicillin-clavulanate; AN, amikacin; AZM, azithromycin; CIP, ciprofloxacin; CLR, clarithromycin; DOX, doxycycline; ENR, enrofloxacin; FOX, cefoxitin; GM, gentamicin; IPM, imipenem; LEV, levofloxacin; LZD, linezolid; MOX, moxifloxacin; SXT, trimethoprim-sulfamethoxazole. Bold values indicate non-susceptibility to antibiotics (amoxicillin-clavulanate may have no efficacy *in vivo* for RGM infection; its susceptibility results are not indicated by bold typeface).
[b]The isolates from dogs included CSME, CMN, and CMFCS. The isolates from cats included FSME, FMCH, and FMFCS. The observations of azithromycin and clarithromycin are presented from a 14-day incubation. CLSI M24, 2018 has not addressed the breakpoints of gentamicin, enrofloxacin, levofloxacin, and azithromycin for RGM. The breakpoints were referred to the interpretation criteria for *Staphylococcus* spp. (CLSI M100 and CLSI VET08).

## DISCUSSION

This study described RGM infections in dogs and cats in Taiwan, focusing on the prevalence of different RGM species and antimicrobial resistance data. Owing to regional differences in RGM species, these data provide more information on the initial selection of antimicrobials for dogs and cats (10, 11). Studies on RGM isolated from these animals in Australia (dogs and cats) (6, 7, 32) have identified *M. smegmatis* as the most common species, followed by *M. fortuitum*. Because *M. abscessus* is less common, fluoroquinolones (moxifloxacin and pradofloxacin) and doxycycline are typically recommended for the

**TABLE 5** MICs, length of erm(41) and point mutation detection of *M. abscessus* complex[a]

| Isolate no. | Subspecies | MIC (µg/mL) | | | | erm(41) (bp) | Position 28 |
|---|---|---|---|---|---|---|---|
| | | CLR | | AZM | | | |
| | | Day 3 | Day 14 | Day 3 | Day 14 | | |
| CMAB-01 | ma | <1 | <1 | 1 | 4 | 246 | T |
| CMAB-05a | ma | <1 | <1 | 1 | 4 | 246 | T |
| CMAB-12 | ma | <1 | <1 | 4 | 8 | 246 | T |
| CMAB-08 | ma | <1 | <1 | 1 | 8 | 246 | T |
| CMAB-02 | ma | <1 | <1 | 2 | 16 | 246 | T |
| CMAB-04 | ma | <1 | <1 | 4 | >16 | 246 | T |
| CMAB-09 | bo | <1 | <1 | 4 | 8 | 522 | C |
| CMAB-03 | ab | <1 | <1 | <1 | 8 | 522 | C |
| **CMAB-10a** | **ab** | <1 | **>16** | <1 | **>16** | **522** | **T** |
| FMAB-04 | ma | <1 | <1 | <1 | 2 | 246 | T |
| FMAB-12 | ma | <1 | <1 | 2 | 2 | 246 | T |
| FMAB-02a | ma | <1 | <1 | 2 | 4 | 246 | T |
| FMAB-01 | ma | <1 | <1 | <1 | 4 | 246 | T |
| FMAB-09 | ma | <1 | <1 | <1 | 8 | 246 | T |
| FMAB-08 | ma | <1 | 1 | 2 | 16 | 246 | T |
| FMAB-11 | bo | <1 | <1 | 2 | 8 | 522 | C |
| **FMAB-05** | **ab** | <1 | **>16** | <1 | **>16** | **522** | **T** |
| **FMAB-10** | **ab** | <1 | **>16** | 4 | **>16** | **522** | **T** |
| **FMAB-13a** | **ab** | <1 | **>16** | 8 | **>16** | **522** | **T** |
| **FMAB-06a** | **ab** | 2 | **>16** | 16 | **>16** | **522** | **T** |

[a] Abbreviations: ab, *M. abscessus*; AZM, azithromycin; bo, *M. bolletii*; CLR, clarithromycin; CMAB, canine *M. abscessus*; FMAB, feline *M. abscessus*; ma, *M. massiliense*. A total of 20 isolates (9 dogs and 11 cats) were tested. The isolates highlighted in bold indicate the presence of functional erythromycin ribosomal methyl transferase (erm(41)) gene.

**TABLE 6** MICs and *erm* detection of *M. fortuitum* complex, *M. nivoides*, *M. smegmatis*, and *M. chelonae*[a]

| Isolate no. | Species | MIC (µg/mL) | | | | *erm* |
|---|---|---|---|---|---|---|
| | | CLR | | AZM | | |
| | | Day 3 | Day 14 | Day 3 | Day 14 | |
| CMFC-01 | *M. fortuitum* | <1 | 2 | 4 | >16 | Negative |
| CMFC-16a | | <1 | 2 | 8 | >16 | Negative |
| CMFC-02 | | 1 | 2 | >16 | >16 | Negative |
| **CMFC-04** | | <1 | **8** | 4 | >16 | **Positive** |
| **CMFC-08a** | | <1 | **>16** | 8 | >16 | **Positive** |
| **CMFC-13a** | | 8 | **>16** | >16 | >16 | **Positive** |
| **CMFC-03** | | >16 | **>16** | 8 | >16 | **Positive** |
| **CMFC-05** | | >16 | **>16** | >16 | >16 | **Positive** |
| **CMFC-06a** | | >16 | **>16** | >16 | >16 | **Positive** |
| **CMFC-10a** | | >16 | **>16** | >16 | >16 | **Positive** |
| **CMFC-15** | | >16 | **>16** | >16 | >16 | **Positive** |
| **FMFC-01e** | | 8 | **>16** | >16 | >16 | **Positive** |
| **FMFC-06** | | 8 | **>16** | >16 | >16 | **Positive** |
| CMFCS-1 | *M. peregrinum* | <1 | <1 | 1 | 4 | Negative |
| **CMFCS-2** | *M. mageritense* | >16 | **>16** | >16 | >16 | **Positive** |
| **FMFCS-1** | | >16 | **>16** | >16 | >16 | **Positive** |
| **CMFCS-3** | *M. porcinum* | <1 | **>16** | 4 | >16 | **Positive** |
| FMFCS-3 | *M. mucogenicum* | <1 | 2 | 16 | >16 | Negative |
| FMFCS-2 | | 1 | 1 | 8 | >16 | Negative |
| FMFCS-4 | *M. houstonense* | 1 | 2 | 8 | >16 | Negative |
| CMN-1 | *M. nivoides* | <1 | <1 | <1 | <1 | Negative |
| CSME-1 | *M. smegmatis* | <1 | 4 | 1 | 8 | Negative |
| **FSME-1** | | <1 | **>16** | 8 | >16 | **Positive** |
| FMCH-1 | *M. chelonae*[b] | <1 | <1 | 8 | 16 | Negative |

[a]AZM, azithromycin; CMFC, canine *M. fortuitum*; CMN, canine *M. nivoides*; CLR, clarithromycin; CSME, canine *M. smegmatis*; FMCH, feline *M. chelonae*; FMFC, feline *M. fortuitum*; FSME, feline *M. smegmatis*. A total of 20 *M. fortuitum* complex isolates (14 dogs and 6 cats), 1 *M. nivoides*, 2 *M. smegmatis*, and 1 *M. chelonae* were tested. The isolates highlighted in bold indicate the presence of erythromycin ribosomal methyl transferase (*erm*) gene.
[b]The *M. chelonae* isolate was negative for the *erm*(55)$^P$, *erm*(55)$^C$, and *erm*(55)$^T$.

treatment of RGM infections in both dogs and cats (10, 11). A previous study on cats in the United States showed similar results to those in Australia (33). However, studies on dogs and cats in the United States have found that *M. fortuitum* and *M. abscessus* are more common than *M. smegmatis* (34, 35), and clarithromycin and fluoroquinolones are more commonly recommended as empirical treatments in these areas (10, 11).

However, RGM infections in dogs and cats have not been studied in Taiwan. We identified only two isolates of *M. smegmatis*, the majority of the isolates being MABC and MFC. MFC (14 out of 25 dogs) was more frequently observed than MABC (9 of 25 dogs) in dogs, whereas in cats, MABC (11 of 19 cats) was more common than MFC (6 out of 19 cats). The distribution of RGM species observed in our study was similar to that reported in studies on dogs and cats in the United States (35), and in human hospitals in Taiwan. Two studies reported that *M. abscessus* (53.8% [168/312] and 46% [92/200]), MFC (32.0% [110/312] and 34.5% [69/200]), and *M. chelonae* (10.9% [34/312] and 19.5% [39/200]) were the most commonly isolated species in human RGM infections (36, 37). MABC is known to exhibit significant resistance to several antimicrobial agents, including doxycycline and fluoroquinolones (1, 8). Since MABC and MFC are the predominant RGM-causing infections in dogs and cats in Taiwan, the use of fluoroquinolones and clarithromycin (or azithromycin) for empirical treatment, instead of doxycycline, is recommended. However, serious MABC infections are refractory to monotherapy with clarithromycin or azithromycin, and there is a risk of developing macrolide resistance (21, 22, 26, 27). Therefore, multidrug therapy should be initiated as soon as possible, based on species identification and *in vitro* susceptibility results.

For humans, the macrolide-containing regimens (a macrolide combined with amikacin, cefoxitin, imipenem, tigecycline, clofazimine, or linezolid) are recommended for serious skin, soft tissue, or pulmonary infections caused by MABC, and the number of drugs used varies based on the presence of inducible macrolide resistance and/or resistance because of 23S rRNA gene mutations (8, 9, 16, 17). While multidrug therapy is also recommended for animals, the details of regimens for MABC infection are unclear, although some reports have mentioned that clarithromycin and linezolid are generally effective (10, 11). This may be due to the limitations associated with the use of these drugs in animals. However, multidrug regimens should be considered in cases that are difficult to treat. In our study, some patients treated only with surgery and clarithromycin (or azithromycin) monotherapy experienced recurrent infections (Table S2), and some patients (4 out of 11 cats and 1 out of 9 dogs) infected with MABC demonstrated inducible macrolide resistance (confirmed by clarithromycin susceptibility testing or detection of the functional *erm*(41) gene). In these patients, multidrug regimens should be considered and administered as early as possible.

Determining inducible macrolide resistance by the 14th day of the susceptibility test is time-consuming. In our study, the presence of *erm* genes was fully correlated with susceptibility to clarithromycin. This indicates that the *erm* genes are the primary mechanism underlying macrolide resistance in MABC and MFC isolates from dogs and cats. The detection of *erm* genes is recommended to predict the susceptibility to macrolides and further assess the potential difficulties in treatment at an early stage. Both azithromycin and clarithromycin are clinically effective against RGM infections (9). However, azithromycin susceptibility testing is not recommended because of its instability in broth microdilutions (38), which explains the inconsistency between azithromycin MIC results and the presence of the *erm* genes. The A2058G/C and A2059G/C point mutations in *rrl* were not detected in any clinical isolates in the present study. Macrolide resistance among the collected isolates was found to not be attributed to the point mutations in the 23S rRNA peptidyltransferase region but could be included in a testing panel alongside *erm* genes to achieve integrated outcomes.

MABC and MFC show poor susceptibility to many antibiotics, making regimens with ≥2 or 3 drugs challenging, particularly in animals, because of limitations in antibiotic use. Amikacin is an appropriate option in multidrug treatment regimens (8–11, 16, 17). Approximately 85.0% of the MABC and MFC isolates were susceptible to amikacin. Two isolates (*M. mageritense*) showed higher amikacin MIC values (128 µg/mL), but no point mutations (positions 1405–1409 and 1491–1496) were found in *rrs* sequencing alignment. Aminoglycoside resistance can develop via various mechanisms. In addition to mutations in *rrs* and *rpsL*, which confer high- or intermediate-level resistance (8, 29, 30, 39), previous studies have suggested that lower levels of resistance may result from other pathways such as the permeability of the cell walls (8, 39). More than 80% of the MABC and MFC isolates in the current study were susceptible to linezolid, a promising option for both MABC and MFC infections (9, 16, 40, 41). Additionally, it has been recommended for use in multidrug treatment regimens against MABC (9, 11, 17); however, its use in animals is restricted.

For MABC, the most common subspecies was *M. massiliense* (6 out of 9 dogs and 6 out of 11 cats), which lacks functional *erm*(41). This suggests that macrolides remain a suitable treatment option and are typically associated with a better outcome. Except for linezolid and cefoxitin, the susceptibility to these drugs was similar to that reported in a study conducted on humans in Taiwan (37). Cefoxitin can be considered an alternative drug in macrolide-containing regimens (9, 16), particularly in untreated cases (8). Clofazimine is another option with potential efficacy in both humans (9) and animals (10, 11, 42); however, this was not evaluated in the present study.

For MFC, therapy with at least two drugs is recommended (8, 9), with fluoroquinolones being an appropriate option. Our MFC isolates showed relatively higher MIC values for other antibiotics compared with the values reported in a previous study in cats in the United States (33), whereas the susceptibility results were similar to those observed

in a previous study conducted in humans in Taiwan (37). Imipenem may be considered an alternative drug; however, its use requires particular caution, as long-term administration in animals is challenging because of the need to minimize its use or reserve it for multidrug-resistant gram-negative bacterial infections (e.g., extended-spectrum beta-lactamase-producing *Enterobacterales*). Although amoxicillin-clavulanate showed *in vitro* activity only against *M. smegmatis*, its clinical effectiveness should not be considered for mycobacterial infections (10).

In the cases included in this study, certain clinical information warrants further attention. Cat 16 (FMAB-11, *M. bolletii*) and Cat 17 (FSME-1, *M. smegmatis*) were diagnosed with lipoid pneumonia, based on histopathology, and they had a history of vomiting and force feeding, which are risk factors for lipoid pneumonia (43–45). *M. abscessus* was isolated from Cat 5 two times from the urine during 2 weeks (FMAB-13a and FMAB-13b, cystocentesis, >$10^3$ cfu/mL growth, pure culture; Tables S1 and S2). In cats without urinary tract symptoms or structural abnormalities, the clinical significance is unclear. The isolation of RGM from urine has been previously described in humans, but the details are lacking (37). Dog 25 (CMFCS-3), with hematuria and prostatic cysts, tested positive for *M. porcinum* in culture from a prostate cyst. Previous studies on bacterial prostatitis in dogs have not reported RGM infections, and their clinical significance remains unclear (10, 11). In some cases of cutaneous pyogranulomas, secondary infections occur during treatment, with *Enterobacterales* as the predominant bacteria (normally with intrinsic resistance to macrolides). Proactive follow-up and intervention can help minimize the impact of treatment regimens. Furthermore, aggressive evaluation to determine the necessity of surgical resection and reconstructive techniques during the treatment process is essential, as such interventions can significantly improve prognosis, particularly in cases of severe infections where the affected tissues may have insufficient concentrations of antimicrobial agents.

To the best of our knowledge, this is the first study of mycobacterial infections in dogs and cats in Taiwan. Given the diagnostic and treatment challenges associated with mycobacterial infections, the data on antibiotic resistance patterns and *erm* genes provided in this study can assist clinical veterinarians. Similar to human treatment guidelines, multidrug regimens are crucial for treating severe mycobacterial infections. However, the use of these essential antibiotics in animals is limited, and finding a balance between clinical efficacy and reducing the use of antibiotics is an important issue to be addressed in the future.

## MATERIALS AND METHODS

### Background

This study was conducted at the Research Center for Animal Medicine, National Chung Hsing University, Taichung, Taiwan. This institution provides clinical microbiology testing services for veterinary hospitals and collects drug-resistant pathogens for further research. None of the details of the isolates in this study contained personal patient or owner information.

### Sample collection

From August 2018 to November 2021, the Research Center for Animal Medicine received 42,433 clinical microbiological specimens from dogs and cats submitted by veterinary hospitals across Taiwan. A total of 44 RGM isolates were identified in 25 dogs and 19 cats. These 44 cases were submitted from 29 veterinary hospitals, with isolation sites including cutaneous infections, respiratory tracts (bronchial lavage and lung tissue), lymph nodes, the urinary tract, and prostate glands. Specimens were collected by clinical veterinarians via fine-needle aspiration biopsy, transport swabs, or fresh tissue collection.

## Bacterial isolation and identification

All specimens were inoculated on 5% blood agar plates (5% sheep blood in tryptic soy agar) and incubated at 35 ± 1°C under ambient air for 7 days. When colonies formed, a single colony was randomly chosen from each specimen and subcultured for purification. Identification of RGM isolates was based on the growth time, colony morphology, Gram's stain results, Ziehl–Neelsen stain results, and sequencing analysis of the 16S rRNA gene and heat shock protein gene (*hsp65*) using the following primer sets: Hsp65 F (5'-ATCGCCAAGGAGATCGAGCT-3') and Hsp65 R (5'-AAGGTGCCGCGGATCTT GTT-3') (46). The identification of *M. abscessus* subspecies (*M. abscessus*, *M. bolletii*, and *M. massiliense*) was performed using a multiplex PCR assay with the following primers: MAB2613F (5'-GTTCGGATCGCATGGCGTTGTGCTG-3'), MAB2613R (5'-GGGATGCTGTGATCG AGGTCGGC-3'), MAB_1655F (5'-GAGGGCACGGGAGAGACCACCGGAG-3'), and MAB_1655R (5'-CCATTTCYCTATCYCGCCCG-3') (47).

## Antimicrobial susceptibility test

The MIC test was performed using the broth microdilution method following the guidelines published by the Clinical and Laboratory Standards Institute (38). Powders of amikacin, gentamicin, cefoxitin, doxycycline, enrofloxacin, ciprofloxacin, levofloxacin, moxifloxacin, azithromycin, clarithromycin, imipenem, linezolid, trimethoprim-sulfame-thoxazole (MedChemExpress, Monmouth Junction, NJ, USA), and amoxicillin-clavula-nate (Cayman Chemical, Ann Arbor, MI, USA) were dissolved in a solution containing the solvent, as previously suggested (48). Inoculated 96-well plates were incubated at 30 ± 0.5°C under ambient air for 3−5 days until visible turbidity was observed, except for macrolides (clarithromycin and azithromycin), which required 14 days of incubation to confirm the presence of inducible resistance (38). Each antimicrobial agent was tested in triplicate. Quality control was performed using *M. peregrinum* ATCC 700686 and *Staphylococcus aureus* ATCC 29213. As the interpretation criteria for gentamicin, enrofloxacin, levofloxacin, azithromycin, and amoxicillin-clavulanate for RGM were absent, we referred to the interpretation criteria for *Staphylococcus* spp. (37, 49, 50).

## Macrolide-resistance gene detection

The reference primer sets CME-1 (5'-ACGTGGTGGTGGGCAAYCTG-3') and CME-2 (5'-AAT TCGAACCACGGCCACCACT-3') were used for *erm* detection in MFC species, *M. nivoides* and *M. smegmatis* (20). The reference primer sets *erm*(41)F (5'-GACCGGGGCCTTCTTCGT GAT-3') and *erm*(41)R1 (5'-GACTTCCCCGCACCGATTCC-3') were used to detect *erm*(41) in MABC isolates (22). To confirm the *erm*(41) deletion and T28C mutation in MABC, Sanger sequencing was conducted using an Applied Biosystems (Waltham, MA, USA) 3730xl DNA Analyzer. The reference primer sets erm(55)[P]-F-1 (5'-CTTGACTGACCAACCGACGA-3'), erm(55)[P]-R-1 (5'-TGTCATGACCCCACCTTTCG-3'), erm(55)[C].2Fa (5'-CAACTACCCTGTTCGCCG TA-3'), erm(55)[C].2Ra (5'-CATCGCCAATTCCTCGAACG-3'), erm(55)[T].3Fb (5'-CCATCGTAGGAAA CCTGCCA-3'), and erm(55)[T].3Rb (5'-CGCGAGGCAAGGATTGATCT-3') were used for *erm*(55) detection in *M. chelonae* (23). The primer sets 23.1 (5'-AATGGCGTAACGACTTCTCAACTGT -3') and 23.2 (5'-GCACTAGAGGTTCGTCCGTCCC-3') were used to amplify *rrl* in all isolates. The products were subjected to bidirectional sequencing to detect point mutations at positions 2058 and 2059 of the 23S rRNA gene (26, 27).

## Sequencing of the *rrs* gene for aminoglycoside resistance

Detection was performed as previously described (30). Reference primer sets rrs1-F (5'-A TGACGTCAAGTCATCATGCC-3') and rrs1-R (5'-AGGTGATCCAGCCGCACCTTC-3') were used for amplification. Sequence alignment was performed to examine the mutations at positions 1405–1409 and 1491–1496 (*Escherichia coli* numbering system).

## ACKNOWLEDGMENTS

The authors would like to express their appreciation to the veterinarians who provided the clinical specimens used in this study.

The project was funded by the Clinical Microbiology Laboratory, Animal Disease Diagnostic Center, National Chung Hsing University.

The authors declare no potential conflicts of interest regarding the research, authorship, and/or publication of this article.

## AUTHOR AFFILIATIONS

[1]Graduate Institute of Veterinary Pathobiology, College of Veterinary Medicine, National Chung Hsing University, Taichung, Taiwan

[2]Research Center for Animal Medicine, National Chung Hsing University, Taichung, Taiwan

[3]Graduate Institute of Molecular and Comparative Pathobiology, School of Veterinary Medicine, National Taiwan University, Taipei, Taiwan

[4]Prestige Veterinary Clinic, Hsinchu, Taiwan

[5]Department of Post-Baccalaureate Veterinary Medicine, College of Medical and Health Science, Asia University, Taichung, Taiwan

[6]Fumao Mansion Animal Hospital, Taichung, Taiwan

[7]Department of Veterinary Medicine, College of Veterinary Medicine, National Chung Hsing University, Taichung, Taiwan

[8]Mercy Animal Medical Center, Kaohsiung, Taiwan

[9]Sincere Veterinary Clinic, New Taipei, Taiwan

## AUTHOR ORCIDs

Shu-Wen Chen  http://orcid.org/0000-0002-0783-445X
Wei-Hsiang Huang  https://orcid.org/0000-0002-0098-0278
Ying-Chen Wu  http://orcid.org/0000-0002-4945-1721

## AUTHOR CONTRIBUTIONS

Shu-Wen Chen, Conceptualization, Data curation, Investigation, Writing – original draft | Ter-Hsin Chen, Resources, Supervision | Wei-Hsiang Huang, Resources | Chia-Chun Hou, Resources | Chen-Jou Lin, Resources | Yi-Fu Chang, software | Hsin-Yi Wu, Resources | Ying-Chen Wu, Conceptualization, Data curation, Funding acquisition, Investigation, Methodology, Supervision, Writing – original draft, Writing – review and editing

## ADDITIONAL FILES

The following material is available online.

### Supplemental Material

**Table S1 (Spectrum03074-24-s0001.xlsx).** Detailed information on all rapidly growing *mycobacteria* isolates (n = 44).

**Table S2 (Spectrum03074-24-s0002.xlsx).** Rapidly growing *mycobacteria* isolates (n = 27) from follow-up examinations of 11 patients.

### Open Peer Review

**PEER REVIEW HISTORY (review-history.pdf).** An accounting of the reviewer comments and feedback.

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
