## [Reviewer comments · Microbiology Spectrum]

Microbiology Spectrum

Antimicrobial resistance of rapidly growing mycobacteria isolated from companion animals in Taiwan

Shu-Wen Chen, Ter-Hsin Chen, Wei-Hsiang Huang, Chia-Chun Hou, Chen-Jou Lin, Yi-Fu Chang, Hsin-Yi Wu, and Ying-Chen Wu

Corresponding Author(s): Ying-Chen Wu, Graduate Institute of Veterinary Pathobiology, College of Veterinary Medicine National Chung Hsing University, Taiwan

Review Timeline:

Submission Date:	November 27, 2024
Editorial Decision:	January 3, 2025
Revision Received:	March 4, 2025
Accepted:	April 3, 2025

Editor: Po-Yu Liu

Reviewer(s): Disclosure of reviewer identity is with reference to reviewer comments included in decision letter(s). The following individuals involved in review of your submission have agreed to reveal their identity: Yung Chun Chen (Reviewer #1)

Transaction Report:

DOI: <https://doi.org/10.1128/spectrum.03074-24>

Re: Spectrum03074-24 (Antimicrobial resistance of rapidly growing mycobacteria isolated from companion animals in Taiwan)

Dear Prof. Ying-Chen Wu:

Thank you for the privilege of reviewing your work. Below you will find my comments, instructions from the Spectrum editorial office, and the reviewer comments.

Revision Guidelines

Sincerely,
Po-Yu Liu
Editor
Microbiology Spectrum

Reviewer #1 (Comments for the Author):

Dear Ying-Chen Wua,

Thank you for submitting your article, which highlights the significance of rapidly growing mycobacteria (RGM) in Taiwan. This is an important and valuable contribution to the field, and I appreciate the effort involved in conducting such challenging work. However, I have some questions and comments that require further clarification:

1. If I understand correctly, this study focused on RGM isolated only from animals (dogs and cats). While I acknowledge the relevance of RGM in human diseases, the connection between the statement "In Taiwan, there is an increasing..." (lines 89-92, page 6) and the preceding sentences is unclear. If evidence of RGM in animals is limited, I suggest adding context to clarify why this evidence is being cited here.
2. In lines 95-96, page 6, you state: "In veterinary hospitals in Taiwan, acid-fast staining is not widely used in cytological examinations." Is there any evidence to support this claim? Providing references or data would strengthen your argument.
3. In line 99, page 6, you describe *M. abscessus* as a "new antibiotic nightmare" due to its multidrug resistance to empirical antibiotics. Please elaborate on what kind of "nightmare" this refers to. For example, you could address the proportion or severity of resistance rather than relying on vague terminology or phrases borrowed from other articles. Additionally, I recommend citing recent references to support your claims.
4. In lines 100-101, page 6, you write: "Macrolides are the major therapeutic agents used against *M. abscessus*." Please consider citing appropriate references to substantiate this statement.
5. In lines 103-106, page 6, you state: "Methylation of the peptidyltransferase region in the 23S subunit by a methyltransferase reduces the affinity between the macrolide and the target region, resulting in inducible resistance, which leads to an increased minimal inhibitory concentration (MIC) after exposure to macrolides." This is a critical point that requires supporting references for accuracy and credibility.
6. In the Results section (line 126 - 131, page 8), were there any duplicates in the samples? If duplicates existed, could you clarify how they were differentiated and included in the analysis? Providing this information would help ensure the transparency and rigor of your methodology.
7. In the Results section, as far as I know, *M. abscessus* belongs to the *M. abscessus* complex, similar to how *M. fortuitum* belongs to the *M. fortuitum* complex. Since you differentiated the *M. fortuitum* complex into *M. fortuitum*, *M. peregrinum*, *M. mageritense*, and *M. porcinum*, could you clarify why only *M. abscessus* was included in your analysis? Does *M. abscessus* here refer specifically to *M. abscessus* within the *M. abscessus* complex, thereby excluding *M. massiliense* and *M. bolletii*? For reference, recent studies have highlighted these distinctions (e.g., *Front Cell Infect Microbiol.* 2021 Apr 26;11:659997. doi: 10.3389/fcimb.2021.659997; *Int J Syst Evol Microbiol.* 2016 Nov;66(11):4471-4479. doi: 10.1099/ijsem.0.001376).
8. In lines 147-155, page 9, please clarify the use of the term "correlation," as no statistical results are provided in this section. Providing further details or justification for this term would enhance the clarity of your argument.
9. In line 156, page 9, please clarify the extent of the "lowerness." Specifically, could you explain how low the values are or the magnitude of the difference? Providing more details would help clarify the significance of this finding. If you would express the comparison, please report the extent or range of differences.
10. In line 169, page 10, please clarify the use of "regionally," as no geographical data are presented in this section.
11. A general comment on your discussion: If resistance is the primary outcome you aim to address, it should be emphasized in the first paragraph of the discussion section, rather than starting with epidemiological patterns and distribution. Another question to consider is whether you can apply your results or provide recommendations for resistance testing in clinical cases.
12. In line 280, page 16, please verify and ensure the correct location of National Chung Hsing University. To the best of my knowledge, this university is located in Taiwan, not China, as indicated in your affiliation. This distinction is essential for maintaining accuracy and clarity in academic writing, as misrepresenting the institution's location could undermine the credibility of the article. However, if the data was indeed retrieved from a school in China, I apologize for the misunderstanding and kindly request that you specify the province to provide accurate context.
13. You mentioned that the sampling was conducted across Taiwan (lines 286-287, page 16). I suggest including these spatial details in Supplemental Table 1S to provide a clearer representation of the sampling locations and to strengthen the epidemiological context of your study.
14. In lines 302-304, page 17, please ensure the version of CLSI guidelines used in your study. Since your research was conducted between 2018 and 2021, I recommend clarifying why an older version of CLSI was used, given that the updated CLSI M24, 3rd edition, was published in November 2018 to replace M24-A2. This clarification is important for ensuring methodological accuracy and relevance.

Reviewer #2 (Comments for the Author):

General comments:

This study investigates the antimicrobial resistance and species distribution of rapidly growing mycobacteria (RGM) in companion animals in Taiwan. A total of 60 RGM isolates were collected from 25 dogs and 19 cats between 2018 and 2021. The dominant species identified were *Mycobacterium abscessus* and *Mycobacterium fortuitum* complex (MFC). This is a first study of rapidly growing mycobacteria from the companion animals in Taiwan. The results provide insights into drug resistance in vet science and information to develop therapeutic strategies against mycobacterial infections in animals. However, the number of strains and detected gene types were limited in this study.

Specific comments:

1. Line107-108: incorrect reference was cited. The gene *erm(41)* was not tested in the reference#14. The suitable reference should be: Nash et al., *Antimicrob Agents Chemother.* 2009 (A Novel Gene, *erm(41)*, Confers Inducible Macrolide Resistance to Clinical Isolates of *Mycobacterium abscessus* but Is Absent from *Mycobacterium chelonae*)
2. Several isolates were from the same patients, which could cause over-counting. Any methods to exclude the identical strains from the same patient, such as PFGE or molecular typing?
3. More resistance markers should be tested, for example *rrl* (constitutive macrolides) and *rrs* (aminoglycosides resistance).
4. *M. chelonae* is one of clinical important RGM in human. Animal-associated *M. chelonae* should also be analyzed.
5. RGM in companion animals showed various and inducible of resistance, and multidrug therapy were suggested. It should be highly interesting to determine the synergistic effects of selected antibiotic combinations (e.g. by time-killing assays or checkerboard assays), and this would be a very good value for the manuscript.

Dear Ying-Chen Wua,

Thank you for submitting your article, which highlights the significance of rapidly growing mycobacteria (RGM) in Taiwan. This is an important and valuable contribution to the field, and I appreciate the effort involved in conducting such challenging work. However, I have some questions and comments that require further clarification:

1. If I understand correctly, this study focused on RGM isolated only from animals (dogs and cats). While I acknowledge the relevance of RGM in human diseases, the connection between the statement "In Taiwan, there is an increasing..." (lines 89–92, page 6) and the preceding sentences is unclear. If evidence of RGM in animals is limited, I suggest adding context to clarify why this evidence is being cited here.

2. In lines 95–96, page 6, you state: "In veterinary hospitals in Taiwan, acid-fast staining is not widely used in cytological examinations." Is there any evidence to support this claim? Providing references or data would strengthen your argument.

3. In line 99, page 6, you describe *M. abscessus* as a "new antibiotic nightmare" due to its multidrug resistance to empirical antibiotics. Please elaborate on what kind of "nightmare" this refers to. For example, you could address the proportion or severity of resistance rather than relying on vague terminology or phrases borrowed from other articles. Additionally, I recommend citing recent references to support your claims.

4. In lines 100–101, page 6, you write: "Macrolides are the major therapeutic agents used against *M. abscessus*." Please consider citing appropriate references to substantiate this statement.

5. In lines 103–106, page 6, you state: "Methylation of the peptidyltransferase region in the 23S subunit by a methyltransferase reduces the affinity between the macrolide and the target region, resulting in inducible resistance, which leads to an increased minimal inhibitory concentration (MIC) after exposure to macrolides." This is a critical point that requires supporting references for accuracy and credibility.

6. In the Results section (line 126 - 131, page 8), were there any duplicates in the samples? If duplicates existed, could you clarify how they were differentiated and included in the analysis? Providing this information would help ensure the

transparency and rigor of your methodology.

7. In the Results section, as far as I know, *M. abscessus* belongs to the *M. abscessus* complex, similar to how *M. fortuitum* belongs to the *M. fortuitum* complex. Since you differentiated the *M. fortuitum* complex into *M. fortuitum*, *M. peregrinum*, *M. mageritense*, and *M. porcinum*, could you clarify why only *M. abscessus* was included in your analysis? Does *M. abscessus* here refer specifically to *M. abscessus* within the *M. abscessus* complex, thereby excluding *M. massiliense* and *M. bolletii*? For reference, recent studies have highlighted these distinctions (e.g., *Front Cell Infect Microbiol.* 2021 Apr 26;11:659997. doi: 10.3389/fcimb.2021.659997; *Int J Syst Evol Microbiol.* 2016 Nov;66(11):4471-4479. doi: 10.1099/ijsem.0.001376).

8. In lines 147–155, page 9, please clarify the use of the term "correlation," as no statistical results are provided in this section. Providing further details or justification for this term would enhance the clarity of your argument.

9. In line 156, page 9, please clarify the extent of the "lowness." Specifically, could you explain how low the values are or the magnitude of the difference? Providing more details would help clarify the significance of this finding. If you would express the comparison, please report the extent or range of differences.

10. In line 169, page 10, please clarify the use of "regionally," as no geographical data are presented in this section.

11. A general comment on your discussion: If resistance is the primary outcome you aim to address, it should be emphasized in the first paragraph of the discussion section, rather than starting with epidemiological patterns and distribution. Another question to consider is whether you can apply your results or provide recommendations for resistance testing in clinical cases.

12. In line 280, page 16, please verify and ensure the correct location of National Chung Hsing University. To the best of my knowledge, this university is located in Taiwan, not China, as indicated in your affiliation. This distinction is essential for maintaining accuracy and clarity in academic writing, as misrepresenting the institution's location could undermine the credibility of the article. However, if the data was indeed retrieved from a school in China, I apologize for the misunderstanding and kindly request that you specify the province to provide accurate context.

13. You mentioned that the sampling was conducted across Taiwan (lines 286–287, page 16). I suggest including these spatial details in Supplemental Table 1S to provide a clearer representation of the sampling locations and to strengthen the epidemiological context of your study.

14. In lines 302–304, page 17, please ensure the version of CLSI guidelines used in your study. Since your research was conducted between 2018 and 2021, I recommend clarifying why an older version of CLSI was used, given that the updated CLSI M24, 3rd edition, was published in November 2018 to replace M24-A2. This clarification is important for ensuring methodological accuracy and relevance.

The manuscript titled “Antimicrobial resistance of rapidly growing mycobacteria isolated from companion animals in Taiwan” investigates the antimicrobial susceptibility and macrolide-resistance genes of 60 rapidly growing mycobacteria (RGM) isolates obtained from 25 dogs and 19 cats. This study provides valuable insights for improving clinical diagnostics and developing multi-drug treatment strategies against RGM infections in companion animals.

Major Concerns:

1. Repeat Sampling and Species Consistency:

The study reports a total of 60 RGM isolates from 44 cases, suggesting that some subjects were infected with multiple RGM species. According to the Materials and Methods section (lines 287-288), repeat sampling was conducted for certain patients during their treatment course. Were the RGMs isolated from multiple samples of the same patient identical in species? If the isolates were of the same species, did they exhibit any differences in antimicrobial sensitivity?

2. Lack of Subspecies Differentiation for *M. abscessus*:

Mycobacterium abscessus comprises three subspecies, but this manuscript does not differentiate between them. Therefore, *Mycobacterium abscessus* complex (MAC) is used throughout the paper to ensure scientific accuracy.

3. MIC Range for *M. fortuitum* in Table 3:

In Table 3, the MIC range for amikacin against *M. fortuitum* is reported as ≤ 4 -128, with the highest MIC value being 128. How can the sensitivity rate be reported as 100% given this broad MIC range? Clarification on how the sensitivity rate was calculated under these conditions is necessary.

4. Discussion vs. Results Section:

The Discussion section should be more concise, focusing on the interpretation of the findings. A more detailed and thorough explanation should be provided in the Results section to offer clearer insights into the data.

Minor points:

1. Line 42: Please use the full name *Mycobacterium abscessus* at its first appearance. For subsequent mentions, the abbreviation "M. abscessus" can be used, as seen in

line 67.

2. Line 71: “our RGM”, line 261 “significance unclear”. line 294 “ $35 \pm 1 \text{ }^\circ\text{C}$ ” have extra spaces.

Reviewer #1 (Comments for the Author):

Thank you for submitting your article, which highlights the significance of rapidly growing mycobacteria (RGM) in Taiwan. This is an important and valuable contribution to the field, and I appreciate the effort involved in conducting such challenging work. However, I have some questions and comments that require further clarification:

Thank you for your guidance and feedback. We appreciate your time and constructive comments.

1. If I understand correctly, this study focused on RGM isolated only from animals (dogs and cats). While I acknowledge the relevance of RGM in human diseases, the connection between the statement "In Taiwan, there is an increasing..." (lines 89-92, page 6) and the preceding sentences is unclear. If evidence of RGM in animals is limited, I suggest adding context to clarify why this evidence is being cited here.

In response to the reviewer's comment, this section was less relevant in the context of the manuscript. I have removed it and added an introduction on MABC (lines 73-77) to bridge the content to the section on macrolides.

2. In lines 95-96, page 6, you state: "In veterinary hospitals in Taiwan, acid-fast staining is not widely used in cytological examinations." Is there any evidence to support this claim? Providing references or data would strengthen your argument.

In response to the reviewer's comment, my argument is limited to my clients in Taiwan (clinical veterinarians) who do not have acid-fast staining systems. Therefore, they commission our lab or other labs to perform the staining when needed. As the reviewer rightly pointed out, the information is limited to my clients (clinical veterinarians) and cannot be presented as scientific facts. Therefore, I have removed that section. Thank you for the reminder.

3. In line 99, page 6, you describe *M. abscessus* as a "new antibiotic nightmare" due to its multidrug resistance to empirical antibiotics. Please elaborate on what kind of "nightmare" this refers to. For example, you could address the proportion or severity of resistance rather than relying on vague terminology or phrases borrowed from other articles. Additionally, I recommend citing recent references to support your claims.

Thank you to the reviewer for the suggestion. I have revised the description to be precise, directly addressing the treatment challenges and resistance issues related to MABC. Additionally, I have included a more recent review article about MABC to support the discussion. (lines 73-77)

4. In lines 100-101, page 6, you write: "Macrolides are the major therapeutic agents used against *M. abscessus*." Please consider citing appropriate references to substantiate this statement.

Thanks to the reviewer's reminder, I have cited four human-related references and two animal-related references: O'Brien et al. (2012, pp. 515-520, in Greene CE (ed.), *Infectious Diseases of the Dog and Cat*, 4th ed., Elsevier, St. Louis, MO) and O'Brien et al. (2023, pp. 739-741, in Sykes JE (ed.), *Greene's Infectious Diseases of the Dog and Cat*, 5th ed., Elsevier, St. Louis, MO). (lines 77)

Unlike the human references, which specify drug combinations for *M. abscessus* (e.g., parenteral treatment with 1–2 drugs from amikacin, imipenem, ceftazidime, or tigecycline, and oral treatment with 2 drugs from azithromycin, linezolid, or clofazimine), the animal references do not provide such detailed regimens.

For animal-related references, O'Brien et al. (2012) note that infections in the U.S. are more appropriately treated with clarithromycin, quinolones, or both. *M. abscessus* is resistant to most antibiotics except clarithromycin and linezolid.

O'Brien et al. (2023) provide fewer details on *M. abscessus* complex treatment. They recommend fluoroquinolones for *M. smegmatis* and *M. fortuitum* infections, clarithromycin or azithromycin for *M. chelonae*, and multidrug therapy (e.g., clofazimine, ceftazidime, linezolid, or amikacin) for refractory cases.

Given these differences in animal references (2012 vs 2023), I have cited both references.

5. In lines 103-106, page 6, you state: "Methylation of the peptidyltransferase region in the 23S subunit by a methyltransferase reduces the affinity between the macrolide and the target region, resulting in inducible resistance, which leads to an increased minimal inhibitory concentration (MIC) after exposure to macrolides." This is a critical point that requires supporting references for accuracy and credibility.

Thank you to the reviewer for the suggestion. I have added references to this statement to enhance its credibility. (lines 81)

6. In the Results section (line 126 - 131, page 8), were there any duplicates in the samples? If duplicates existed, could you clarify how they were differentiated and included in the analysis? Providing this information would help ensure the transparency and rigor of your methodology.

Thank you for the reviewer's inquiry. In this study, all 44 patients were distinct individuals, with no duplicate cases. Among these 44 patients, 11 were more willing to undergo follow-up examinations during treatment, resulting in more than one isolate per patient in these 11 cases. Ultimately, a total of 60 isolates were obtained from the 44 patients.

As suggested by the reviewer, to avoid overcounting and to more accurately reflect species distribution, the results have been revised to report patient numbers (patients = 44, isolates = 44), rather than the total number of isolates, including those from follow-up examinations isolates (this has been corrected in the text and Tables 1 to 6).

The isolates from the 11 patients who underwent follow-up examinations are presented separately in Table S2. Since only a minority of patients were available for long-term follow-up, we aimed to extract useful data from this subset, such as changes in antimicrobial resistance, *erm* gene, *rhl* gene, *rrs* gene, and secondary infections during treatment.

Therefore, we conducted a comprehensive analysis of all isolates. Overall, there were minimal changes in antimicrobial resistance, inducible resistance to clarithromycin, and secondary infections in the follow-up isolates. However, in one cat (*M. fortuitum*, strain FMFC-1), resistance to fluoroquinolones increased during treatment. Between the first and second sampling, a point mutation occurred at position 83 of the DNA gyrase A (*gyrA*) gene (initial culture: Ser-83; follow-up: Ser-83 to Trp). This result has been added to Table S2. In this study, we performed *gyrA* gene mutation analysis at position 83 for all isolates. However, many results were difficult to interpret, as the point mutations did not fully correlate with antimicrobial resistance outcomes. Currently, we plan to conduct further analysis on these data in the future. Therefore, we decided not to include the *gyrA* mutation data in this manuscript.

7. In the Results section, as far as I know, *M. abscessus* belongs to the *M. abscessus* complex, similar to how *M. fortuitum* belongs to the *M. fortuitum* complex. Since you differentiated the *M. fortuitum* complex into *M. fortuitum*, *M. peregrinum*, *M. mageritense*, and *M. porcinum*, could you clarify why only *M. abscessus* was included in your analysis? Does *M. abscessus* here refer specifically to *M. abscessus* within the *M. abscessus* complex, thereby excluding *M. massiliense* and *M. bolletii*? For reference, recent studies have highlighted these distinctions (e.g., Front Cell Infect Microbiol. 2021 Apr 26;11:659997. doi: 10.3389/fcimb.2021.659997; Int J Syst Evol Microbiol. 2016 Nov;66(11):4471-4479. doi: 10.1099/ijsem.0.001376).

Thank you for providing your professional insights. The original manuscript did not include subspecies differentiation of *M. abscessus*, as we had not found a suitable method for distinguishing them at the time. In the last month, we identified an appropriate method (Yoshida M, Sano S, Chien JY, Fukano H, Suzuki M, Asakura T, Morimoto K, Murase Y, Miyamoto S, Kurashima A, Hasegawa N, Hsueh PR, Mitarai S, Ato M, Hoshino Y. 2021. A novel DNA chromatography method to discriminate *Mycobacterium abscessus* subspecies and macrolide susceptibility. EBioMedicine 64:103187) and expanded our analysis to include the *M. abscessus* complex subspecies (*M. abscessus*, *M. massiliense*, and *M. bolletii*), which has been incorporated into Tables 1, 2, 5, and the manuscript, to providing additional information.

8. In lines 147-155, page 9, please clarify the use of the term "correlation," as no statistical results are provided in this section. Providing further details or justification for this term would enhance the clarity of your argument.

Thank you for providing your professional insights. The confused terms have been revised to describe the values. The modified paragraph is: **(lines 124-127)**

9. In line 156, page 9, please clarify the extent of the "lowerness." Specifically, could you explain how low the values are or the magnitude of the difference? Providing more details would help clarify the significance of this finding. If you would express the comparison, please report the extent or range of differences.

Thank you for the reviewer's suggestion. The detailed data has been presented, and additional descriptions have been added. The modified paragraph is: **(lines 128-129)**

10. In line 169, page 10, please clarify the use of "regionally," as no geographical data are presented in this section.

This paragraph intends to describe that studies from Australia and the U.S. are more abundant in the references regarding RGM infections in dogs and cats. In Australia, there are more cases of *M. smegmatis*, while in the U.S., both *M. fortuitum*, *M. abscessus*, and *M. smegmatis* are prevalent. The differences in the prevalent species determine the use of empirical therapy. This paragraph has been revised to provide a more detailed description, which includes: **(lines 143-152)**

11. A general comment on your discussion: If resistance is the primary outcome you aim to address, it should be emphasized in the first paragraph of the discussion section, rather than starting with epidemiological patterns and distribution. Another question to consider is whether you can apply your results or provide recommendations for resistance testing in clinical cases.

Thank you for the suggestion. I have now merged the discussion to include antimicrobial resistance, epidemiological patterns, and distribution. **(lines 142-152)**

The reason for emphasizing epidemiological patterns and distribution is because these factors are directly related to the choice of treatment. In the references regarding RGM in dogs and cats, a main focus is the predominant species in Australia, *M. smegmatis*, makes doxycycline and fluoroquinolones suitable for empirical treatment. In contrast, in the U.S., where *M. smegmatis* is not the main species, fluoroquinolones and clarithromycin are more appropriate choices for empirical therapy.

In this study, the predominant RGM species in dogs and cats in Taiwan are MFC and MABC. The recommendation for doxycycline use is relatively low, and fluoroquinolones and clarithromycin are suggested as empirical treatments. After identifying the species and antimicrobial resistance (with macrolide-induced resistance being assisted by resistance gene testing), the treatment regimen can be adjusted based on susceptibility data to the appropriate drug combination.

Based on our current data, we suggest that veterinary clinicians consider complete susceptibility testing (regardless of species, which must include inducible macrolide resistance) when RGM infection is suspected. If there are time constraints, such as in cases of pulmonary Mycobacterial infections, Lab can perform PCR testing for both *hsp65* (for identification of *Mycobacterium*) and 16S rRNA (to confirm the presence of one or more bacteria in the affected area) directly on the

sample (if only acid-fast bacilli are observed on the smear) or on a culture after 2 to 3 days (there may be colonies that are not visible to the naked eye, but PCR can detect them). Sequence analysis can be used to preliminarily identify the species, which in Taiwan (dogs and cats), in the similar differential diagnosis, may include RGM (MABC, MFC, *M. smegmatis*, *M. chelonae*), *M. avium* complex, or others. The therapy should then be adjusted based on the results of species identification and resistance gene testing. If the case is expected to be difficult to treat, multidrug therapy (> 2 drugs) should be initiated from the start.

Regarding recommendations for clinical medication, this study discusses the differences between human and animal treatments and highlights variations in antibiotics between Australia and the United States. In the initial treatment phase, empirical drug selection in Taiwan often refers to U.S. based findings (fluoroquinolones and clarithromycin). Once the RGM species is identified and susceptibility data are available, human references can be consulted, as it provides more in-depth exploration of treatment options, including recommendations for various RGM species, the number of drugs in multidrug therapy, and treatment duration (e.g., initial phase or continuation phase). These insights can serve as a reference for animal treatment.

However, determining which drugs are suitable for combined use in animal RGM infections remains challenging. Based on the experiences of clinical veterinarians collaborating with our laboratory, the use of second-line drugs (e.g., linezolid, imipenem) in animals is relatively limited, while others, such as amikacin, are less frequently used due to side effects. Additionally, treatment durations typically range from 3 to 12 months, making it challenging to fully adopt human guidelines. Currently, it is difficult to collect comprehensive clinical case data for further investigation. Consequently, this study could not fully address the clinical responses to various drug combinations. In cases of localized cutaneous infection (pyogranulomatous), an important concept is to initially control the infected area with antibiotics and then gradually assess the necessity of surgical resection. This aspect has been added to the manuscript. **(lines 232-235)**

12. In line 280, page 16, please verify and ensure the correct location of National Chung Hsing University. To the best of my knowledge, this university is located in Taiwan, not China, as indicated in your affiliation. This distinction is essential for maintaining accuracy and clarity in academic writing, as misrepresenting the institution's location could undermine the credibility of the article. However, if the data was indeed retrieved from a school in China, I apologize for the misunderstanding and kindly request that you specify the province to provide accurate context. Thank you for the reviewer's assistance, and I sincerely apologize for the confusion. This was my mistake. The error occurred after I submitted the manuscript for English editing, where the editing team mistakenly added 'China' after 'National Chung Hsing University. I failed to catch this mistake. I have now thoroughly reviewed the manuscript, and there are no further errors.

13. You mentioned that the sampling was conducted across Taiwan (lines 286-287, page 16). I

suggest including these spatial details in Supplemental Table 1S to provide a clearer representation of the sampling locations and to strengthen the epidemiological context of your study.

Thank you for the reviewer's suggestion. We have added the region corresponding to each patient in table S1, along with additional detailed information.

14. In lines 302-304, page 17, please ensure the version of CLSI guidelines used in your study. Since your research was conducted between 2018 and 2021, I recommend clarifying why an older version of CLSI was used, given that the updated CLSI M24, 3rd edition, was published in November 2018 to replace M24-A2. This clarification is important for ensuring methodological accuracy and relevance.

Thank you for the reviewer's reminder. We have updated our analysis to follow CLSI M24, 2018 (CLSI. 2018. Susceptibility Testing of Mycobacteria, Nocardia spp., and Other Aerobic Actinomycetes, 3rd ed. CLSI standard M24. Clinical and Laboratory Standards Institute. Wayne, PA.), and interpreted the results according to the new standards. The manuscript and references have been revised accordingly.

Reviewer #2 (Comments for the Author):

General comments:

This study investigates the antimicrobial resistance and species distribution of rapidly growing mycobacteria (RGM) in companion animals in Taiwan. A total of 60 RGM isolates were collected from 25 dogs and 19 cats between 2018 and 2021. The dominant species identified were *Mycobacterium abscessus* and *Mycobacterium fortuitum* complex (MFC). This is a first study of rapidly growing mycobacteria from the companion animals in Taiwan. The results provide insights into drug resistance in vet science and information to develop therapeutic strategies against mycobacterial infections in animals. However, the number of strains and detected gene types were limited in this study.

Thank you for providing your professional insights. The proportion of RGM infections from dogs and cats in our laboratory is relatively low, resulting in a limited total number of collected cases. Following the reviewers' suggestions. In the last month, we have expanded our analysis to include additional genotypic and resistance gene testing to enhance the information provided in this study. Specifically, we have further classified the *M. abscessus* complex into subspecies (*M. abscessus*, *M. massiliense*, and *M. bolletii*), which have been incorporated into Table 1, Table 2, and the manuscript. Additionally, we have tested for aminoglycoside resistance genes (now included in the manuscript and table S1). These additions will be addressed in the responses below.

Specific comments:

1. Line107-108: incorrect reference was cited. The gene *erm(41)* was not tested in the reference#14. The suitable reference should be: Nash et al., Antimicrob Agents Chemother. 2009 (A Novel Gene, *erm(41)*, Confers Inducible Macrolide Resistance to Clinical Isolates of *Mycobacterium abscessus* but Is Absent from *Mycobacterium chelonae*)

Thank you to the reviewer for your assistance. The references have been revised, and the correct citations have been made. (lines 81-88)

2. Several isolates were from the same patients, which could cause over-counting. Any methods to exclude the identical strains from the same patient, such as PFGE or molecular typing?

Thank you for the reviewer's inquiry. In this study, all 44 patients were distinct individuals, with no duplicate cases.

Among these 44 patients, 11 were more willing to undergo follow-up examinations during treatment, resulting in more than one isolate per patient in these 11 cases. Ultimately, a total of 60 isolates were obtained from the 44 patients.

As suggested by the reviewer, to avoid overcounting and to more accurately reflect species distribution, the results have been revised to report patient numbers (patients = 44, isolates = 44), rather than the total number of isolates, including those from follow-up examinations isolates (this has been corrected in the text and Tables 1 to 6).

The isolates from the 11 patients who underwent follow-up examinations are presented separately in Table 2S. Since only a minority of patients were available for long-term follow-up, we aimed to extract useful data from this subset, such as changes in antimicrobial resistance, *erm* gene, *rhl* gene, *rrs* gene, and secondary infections during treatment.

Therefore, we conducted a comprehensive analysis of all isolates. Overall, there were minimal changes in antimicrobial resistance, inducible resistance to clarithromycin, and secondary infections in the follow-up isolates. However, in one cat (*M. fortuitum*, strain FMFC-1), resistance to fluoroquinolones increased during treatment. Between the first and second sampling, a point mutation occurred at position 83 of the DNA gyrase A (*gyrA*) gene (initial culture: Ser-83; follow-up: Ser-83 to Trp). This result has been added to Table S2.

In this study, we performed *gyrA* gene mutation analysis at position 83 for all isolates. However, many results were difficult to interpret, as the point mutations did not fully correlate with antimicrobial resistance outcomes. Currently, we plan to conduct further analysis on these data in the future. Therefore, we decided not to include the *gyrA* mutation data in this manuscript.

Currently, we are unable to perform further molecular typing (such as PFGE) to confirm whether these isolates are identical. However, given that their antimicrobial resistance profiles remained largely unchanged compared to the initial isolates, it is highly likely that molecular typing would indicate the same isolates.

3. More resistance markers should be tested, for example *rrl* (constitutive macrolides) and *rrs* (aminoglycosides resistance).

Thank you for the reviewer's suggestion. For *rrl*, this study performed testing; however, no point mutations were detected in any of the isolates, so this result was not emphasized. The description of *rrl* results can be found in the text on **(lines 136-137)**

Following your suggestion, the *rrs* testing was conducted last month, but no point mutations were observed in any of the isolates. The results have been added to the text **(lines 136-137)** and Tables S1 and S2.

4. *M. chelonae* is one of clinical important RGM in human. Animal-associated *M. chelonae* should also be analyzed.

Thank you for the reviewer's suggestion. The samples in this study were collected from clinical cases. Among the 44 patients included in the study results, only one case of *M. chelonae* (cat) was identified, and the analysis results (*erm(55)*) were completed last month and have been incorporated into the text **(lines 129-130, 284-288)** and **Table 6**. Since the completion of the study, only one additional case of *M. chelonae* has been received by the laboratory from 2023 to 2025 (on average, the laboratory encounters 1-2 cases of RGM infections per month). If we receive a sufficient number of cases in the future, we will conduct a separate analysis.

5. RGM in companion animals showed various and inducible of resistance, and multidrug therapy were suggested. It should be highly interesting to determine the synergistic effects of selected antibiotic combinations (e.g. by time-killing assays or checkerboard assays), and this would be a very good value for the manuscript.

Thank you for the reviewer's suggestion. We also agree that the study of synergistic effects is important due to the necessity of multidrug therapy. We have considered this, but the complexity and scope of the research are quite large. Currently, we plan to treat it as a separate research project in the future.

Re: Spectrum03074-24R1 (Antimicrobial resistance of rapidly growing mycobacteria isolated from companion animals in Taiwan)

Dear Prof. Ying-Chen Wu:

Your manuscript has been accepted, and I am forwarding it to the ASM production staff for publication. Your paper will first be checked to make sure all elements meet the technical requirements. ASM staff will contact you if anything needs to be revised before copyediting and production can begin. Otherwise, you will be notified when your proofs are ready to be viewed.

Sincerely,
Po-Yu Liu
Editor
Microbiology Spectrum

Reviewer #2 (Comments for the Author):

Minor comments: Should add the subject number for the animals in Table S1.

Reviewer #3 (Comments for the Author):

No